# The Performance of Microfiltration Using Hydrophilic and Hydrophobic Membranes for Phenol Extraction from a Water Solution

Tamara Kawther Hussein [1], Nidaa Adil Jasim [2] and Abdul-Sahib T. Al-Madhhachi [3,*]

1   Environmental Engineering Department, College of Engineering, Mustansiriyah University, Baghdad 10047, Iraq; tamarahussein@uomustansiriyah.edu.iq
2   Highway and Transportation Engineering Department, College of Engineering, Mustansiriyah University, Baghdad 10047, Iraq; nidaa.albayati@uomustansiriyah.edu.iq
3   Water Resources Engineering Department, Mustansiriyah University, Baghdad 10047, Iraq
*   Correspondence: abdu@okstate.edu or a.t.almadhhachi@uomustansiriyah.edu.iq

**Abstract:** Two types of membranes, for hydrophilic and hydrophobic microfiltration, were prepared as flat sheets to treat a phenol-contaminated water solution. The membranes were fabricated using four synthetic polymers: polysulfone, polyethylene oxide, dimethylacetamide, and N-methyl-2-pyrrolidone. Scanning electron microscope measurements of the top-surface and cross-section images of the produced membranes were used to characterize them physically. Distilled water and water contaminated with phenol were used to evaluate the membrane's performance based on the flux results depending on pressure, the concentration of phenol, and temperature variables. Meanwhile, the rejection performance was evaluated using the phenol-contaminated water solution. The results show that the flux increased with increases in pressure and temperature and decreased with increases in phenol concentration. Distilled water gave far higher results than water contaminated with phenol. The flux of distilled water ranged from 52.18 to 73.15 $L/m^2/h$ for the hydrophilic type and from 72.27 to 97.46 $L/m^2/h$ for the hydrophobic type, whereas the flux of water contaminated with phenol solution ranged from 26.58 to 61.55 $L/m^2/h$ for the hydrophilic type and from 29.98 to 80.55 $L/m^2/h$ for the hydrophobic type. Meanwhile, the phenol solution's rejection was 60% when using a hydrophilic membrane, whereas it was only 45% when a hydrophobic membrane was used. The hydrophobic membrane showed high fluxes and low rejection. Thus, transport through this membrane is closer to having viscous behavior than that through the hydrophilic membrane; in contrast, the permeability through the hydrophilic membrane is less because the pore size decreases the viscous flow mechanism.

**Keywords:** membranes; phenol solution; microfiltration; rejection; flux; polysulphone

## 1. Introduction

The removal of phenol is of great interest in wastewater treatment. Phenol production is 8 million tons per year worldwide [1], because phenol is one of the most important intermediates in the chemical industry, particularly for manufacturers of pharmaceuticals, petrol, and iron [2]. With the development of industry, phenolic resin adhesive has come to be used in plywood, wood processing, automobiles, and composite materials [3]. On the other hand, studies confirm that phenol is one of the most harmful pollutants when it seeps from factories and is mixed with surface water. The contamination of runoff and land with phenol poses a threat to human life, flora, and fauna [4].

Depending on the actual succession of scientific studies, oxidation and flotation [5] may have been one of the first methods used to remove phenol, followed by the chemical coagulation method [6] and then the adsorption method using chemical or natural materials [7,8]. All the aforementioned methods have efficient results and/or are economical and

easy to manage. Scientific progress and continuous research led to another method for treating wastewater that contains phenol with remarkable efficiency using a filter containing one ingredient and one layer [9] or two different ingredients [10].

Membrane technologies are dependable and economically practical. They have advantages such as low power consumption, high-quality effluent, and easy scaling up with membrane modules [11–13]. A membrane with very fine pores can remove individual molecules less than 0.0001 μm in size [14]. Using ultrafiltration, membranes made of polymers can remove particles ranging from 0.03 to 0.1 μm in size from industrial wastewater [15]. When additives are used, the membrane's performance can be increased; e.g., polyethylene glycol (PEG) and acetone were used to enhance the permeation of fluid through a membrane [16]. However, the fouling problem limits the usage of ultrafiltration membranes, in spite of the modification of polyethersulfone (PES) ultrafiltration membranes by the blending of O-carboxymethyl chitosan and $Fe_3O_4$ nanoparticles in a PES solution [17]. However, with the usage of a micro-filtration membrane (between 0.1 and 10 μm), it was found that the use of additives such as PEG and polyvinylpyrollidone (PVP) reduces membrane fouling. Additives increase the efficiency of the membrane by changing its properties and improve it by increasing the size of the pores, which reduces the percentage of fouling [18]. A membrane was efficient in removing phenol from industrial wastewater when using OP-4 as a surfactant and kerosene as a solvent in the optimum conditions (the concentration of surfactant OP-4 in the organic membrane phase, the chemical ratio, the concentration of alkali in the solution, and the volume ratio of the organic membrane phase to the internal phase) [19]. A hybrid membrane may be a plausible method for removing phenol and is considered an additional step to be added to the basic steps of treating wastewater containing phenol. The latter method may lead to greater removal efficiency, reduce the rotting obstacle, and save energy [20]. There are also membranes that have been used to remove phenol and are manufactured from cellulose triacetate and cellulose acetate. A type of membrane that uses cellulose materials with high efficiency in acidic solutions was used to treat wastewater contaminated with organic materials [21].

One of the most common materials used in the manufacturing of membranes is polysulfone. In Iraq, there are many petroleum-refining industries, and phenol is one of their most dangerous waste products. Therefore, the importance of the study is to remove phenol using a membrane made of a polysulfone polymer. Hydrophilic and hydrophobic membranes were made by a phase-inversion technique using polysulfone (PSF) as a polymer and dimethylacetamide (DMAc) as a solvent for hydrophobic membranes; and PSF and polyethylene oxide (PEO) as additives and N-methyl-2-pyrrolidone (NMP) as a solvent for hydrophilic membranes. The impacts of operational factors (pressure, concentration, and temperature) are elaborated to cover the area of optimal results. Additionally, membrane permeation was calculated in terms of flux and solute rejection. The morphology of each membrane was analyzed using scanning electron microscopy.

## 2. Materials and Methods

### 2.1. Feed Solutions

In this study, the distilled water (conductivity, 0.22 μs/cm) was used as a feed solution. The phenol ($C_6H_6O$) was purchased from a local supplier at Baghdad, Iraq, and it was prepared in the laboratory by dissolving one gram of phenol crystals in 1 L of distilled water. The mixture was agitated at a speed of 2000 rpm for about 10 min using a magnetic stirrer (LMS-1003, DAIHAN LAB TECH, 0–2000 rpm, Namyangju, Republic of Korea). The phenol dissolved totally to prepare a stock solution of 1000 mg/L concentration and was then diluted to the desired solution concentrations (10, 50, 80 mg/L). The chemical specifications of phenol crystals are listed in Table 1.

**Table 1.** Chemical specifications of phenol crystals.

| Chemical Name | Phenol |
|---|---|
| Formula | $C_6H_6O$ |
| Appearance | White crystalline solid |
| Molecular weight | 94.11 g/mole |
| Solubility in water | 8.2 g/100 mL $H_2O$ |
| Octanol-Water (Log ($K_{ow}$)) | 1.5 |
| Specific gravity | 1.058 |
| Manufacturing company | BDH, England |
| Purity (%) | 99.5 |

*2.2. Membrane*

The manufactured membrane materials were polysulphone, dimethylacetamide, polyethylene oxide, and N-methyl-2-pyrrolidone. PSF is a chemical-resistant material, even if the solution in which it was placed is acidic or basic. DMAc was used as a solvent (with a hydrophobic membrane). The PEO was used as an additive. The NMP was mixed with polysulfone as a solvent for the hydrophilic membranes.

The work for the membrane production took several steps. Firstly, to prepare the hydrophobic membrane, PSF and DMAc were purchased from the market. About 18% PSF and 82% DMAc, by weight, were mixed in a beaker that contained 500 mL of distilled water. DMAc was added first, since it is considered a solvent for the PSF, and later, the PSF was added gradually. To induce dissolution, the liquid was heated to 100 °C and continuously stirred. The stirring duration was 6 h, and it was conducted by a magnetic stirrer. After the mixing process, it was cooled for 15 h. The previous cooling process was critical in releasing the formed bubbles during mixing. If the bubbles were not released, then the size of the pores inside the membrane mixture would result in a poor-quality membrane. The solution was cast on a glass plate with a casting knife with a gap of 400 mm at a manually constant casting speed. The mixture was spread on the glass plate within 10 s in order to avoid chances of thickening the mixture and forming white membranes, which were placed overnight in a water bath at 25 °C. The resulting film was placed on an aluminum slice and dried in a hot place with a temperature less than 65 °C.

Secondly, for the hydrophilic membrane, the preparation was similar to that used in preparing the hydrophobic membrane, with the following differences: In order to improve the membrane's properties and make it hydrophilic, PEO was added, and NMP was added as a solvent. The aforementioned substances were mixed by weight: 18% PSF, 22% PEO, and 60% NMP. Five-hundred milliliters of distilled water was prepared in a beaker, and the solvent was placed in it; afterwards, the PSF was added gradually while the solution was heated up during the mixing process to 100 °C. After the stirrer completely mixed the solution, the mixture was cooled to a temperature of 30 °C. To ensure homogeneity, the PEO was added to the mixture and heated for 30 min. The whole mixing process took 4–6 h. Following the casting process, the glass plate was cautiously immersed in the water bath while NMP diffused out of the membrane sheet. However, water diffused into the membrane for 2 min. To ensure that the phase-separation process was completed, the membrane was stored in a water bath for 12 h at 90 °C. Finally, the membrane was washed out with distilled water several times before it was ready for testing.

*2.3. Experimental Work*

A microfiltration process was tested in a bench-scale system, as shown in Figure 1. The system consists of a glass feed tank with a capacity of 20 L where the feed solution is placed. The feed solution was pumped by a centrifugal pump (11.41–54.50 L/min flow rate, 210 watt power, and manufactured by Stuart Turner Ltd., Henley-on-themes, England, from the feed tank to membrane cells. Membrane cells contain a circular flat sheet membrane (hydrophobic or hydrophilic membrane) with a diameter of 5.75 cm and an effective membrane area of 26 cm². The desired temperature for the feed solution

was varied (10–50 °C), which was controlled by the submersible electrical coil (220 volts, 1000 watts) and the thermostat within the range of 0–80 °C. The water flow rate was regulated by a valve and measured by a rotameter (10–100 L/h); all experiments were carried out at a flow rate of 30 L/h. To measure the desired feed solution pressures of 1.0, 1.5, and 2.0 bars, a pressure gauge (with a range of 0–3 bar) was used. Permeate volume (filtered water) was collected every 10 min in a 50 mL cylinder to calculate water flux; the duration for each experiment was 60 min. After each experiment, the system contaminated with phenol was washed out with distilled water for 30 min at 1.5 bar. The content of phenol was determined by the Folin–Ciocalteu method as described in Jasim and Hussein's [22] study. The absorbance was measured by a UV spectrophotometer (Shimadzu 1800, Kyoto, Japan) at a wavelength of 270 nm. The absorbance recorded from the UV spectrophotometer was used to prepare a standard calibration curve to determine phenol permeate concentration. The concentration of phenol in the permeate was measured by a UV spectrophotometer (Shimadzu 1800, Kyoto, Japan) at a wavelength of 270 nm. The phenol rejection percentage (*R*%) is calculated by the following equation [23,24]:

$$R(\%) = \frac{(C_f - C_p)}{C_f} \times 100 \tag{1}$$

where $C_f$ is the concentration of phenol in a feed solution (mg/L) and $C_p$ the concentration of phenol in permeate (mg/L).

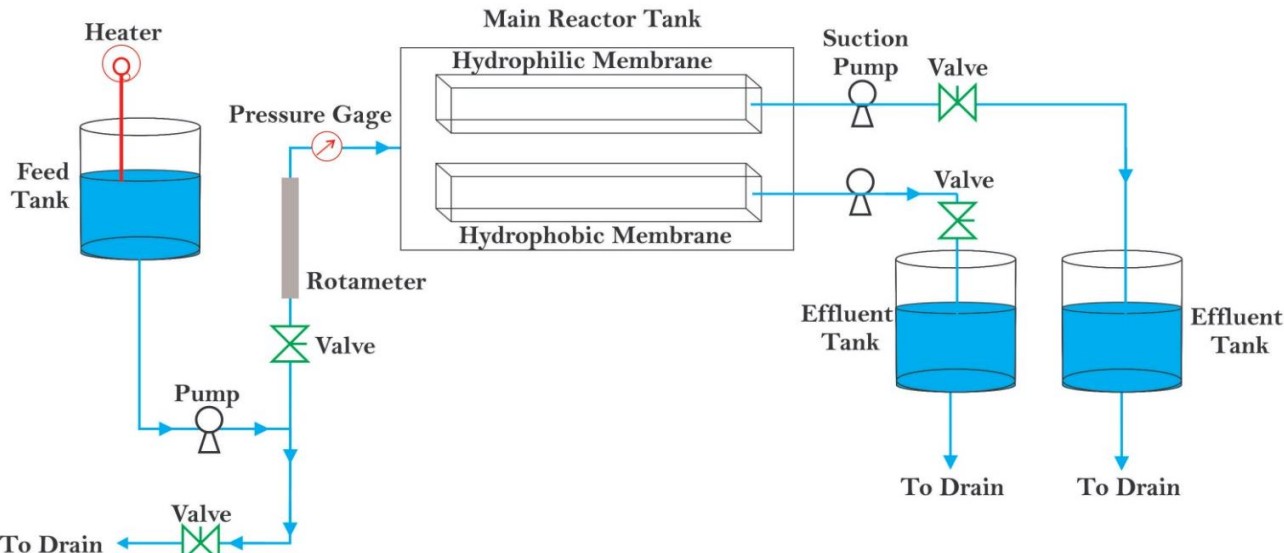

**Figure 1.** Schematic diagram of the microfiltration process.

The permeate flux is calculated by the following equation [18]:

$$J_w = \frac{V}{\Delta t A} \tag{2}$$

where $J_w$ is water flux (L/m$^2$ h), $V$ is permeated volume (L), $\Delta t$ is the sampling time (h), and $A$ is the membrane area (m$^2$). The surface morphology of the membranes was studied by a scanning electron microscope (SEM) model (FEI-USA) after submerging the membrane material in liquid nitrogen and coating it with gold.

## 3. Results and Discussion

### 3.1. Membrane Characterization

Figures 2 and 3 show the top surfaces and cross-sectional SEM micrographs of the hydrophobic and hydrophilic membranes, respectively. Both membrane's general structure consisted of a dense skin layer on top and a porous support sublayer. Figure 2a,b show the top surface and cross-section morphology, respectively, of the hydrophobic membrane. The membrane had a spongy, dense structure, and a few separated closed ends made it look like a drop membrane. The cross-section having large pores is because of the DMAc solvent being used without additives, and on the outer surface, one could note the presence of a few tiny closed pores. During the phase inversion process, the formation of the voids could have been due to the penetration of the solution through the membrane's surface. With the addition of PEO, the main characteristics of the membrane (through the cross-section) were a thick layer in the upper part and a pore in the lower part that looked like a finger (instead of the drop-like membrane, i.e., the hydrophobic membrane). The spongey parts still existed; as a result, the membrane has low resistance to water penetration, as shown in Figure 3a,b. Researchers have indicated that a finger-shaped section's solution-penetration efficiency is better than that of a spongey section [15,17].

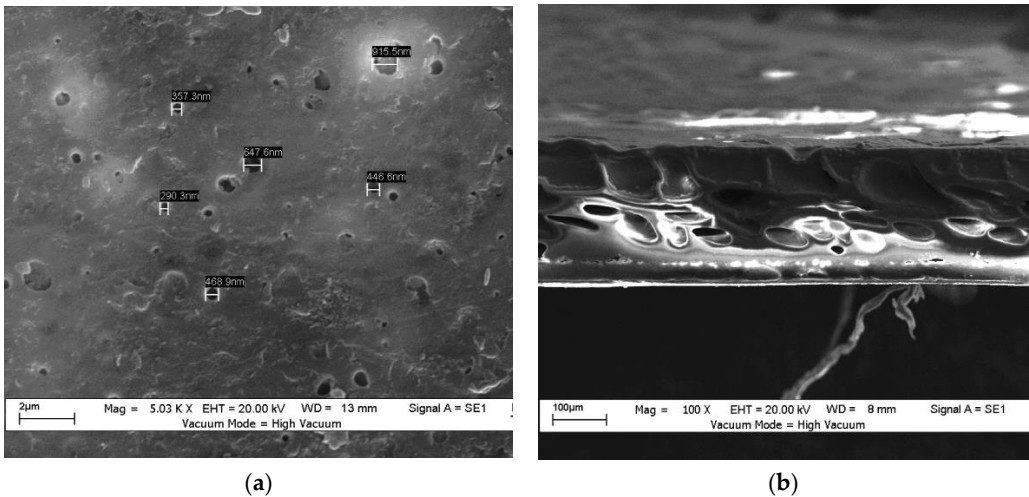

| (a) | (b) |

**Figure 2.** SEM images: (**a**) top surface of the hydrophobic membrane, and (**b**) cross-section of the hydrophobic membrane.

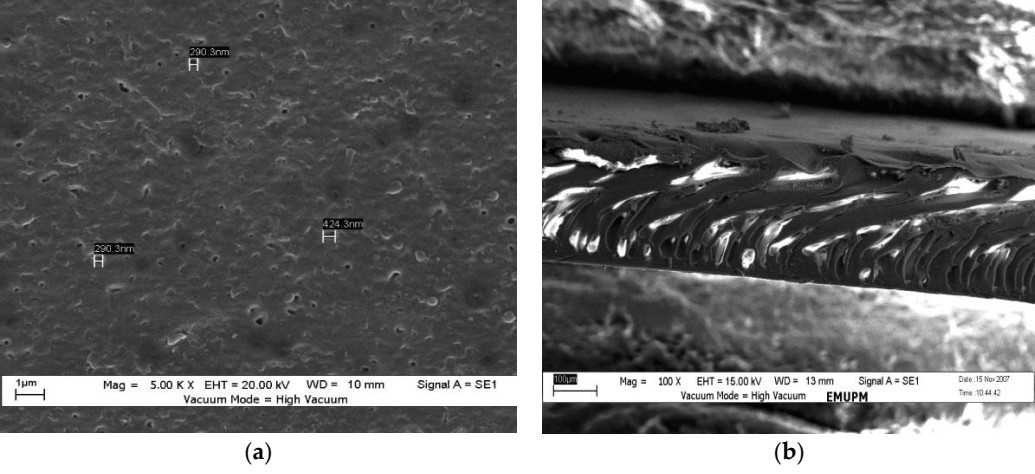

| (a) | (b) |

**Figure 3.** SEM images: (**a**) top surface of the hydrophilic membrane and (**b**) cross-section of the hydrophilic membrane.

As shown in Figures 2a and 3a, it is clear that the hydrophobic membrane had larger pores than the hydrophilic membrane, and this is because the hydrophobic membrane did not contain PEO as an additive. Moreover, PEO increases the viscosity, which gave the membrane a hydrophilic feature. However, increasing the amount of NMP will increase the pore size. NMP, on the other hand, is a powerful solvent for PSF, thereby lowering the viscosity of the solution and increasing the spread rate of water and NMP out of the thin film casting; as a result, the polymer concentration at the water-casting film's contact surface was lower, and the membrane sheet's porosity was higher.

SEM was used to determine the average pore size and pore-size distribution. The top surface, the bottom surface, and a section of the cross-surface membrane were fractured cryogenically in liquid nitrogen to leave an un-deformed structure. Then, they were attached to sample stubs with double-surface gold using a sputter coater. After that, the samples were imaged using SEM. The image analysis was carried out to obtain the average pore size. The average pore size was 521 nm for the hydrophobic membrane (Figure 2a). The average pore size for the hydrophilic membrane was 335 nm; see Figure 3a.

The mechanical stability of the membranes was assessed by the determination of the Young's moduli and elongation at break values of the prepared membranes. It is essential to study the mechanical stability to evaluate the lifetime of a membrane, which could be obtained by examining the mechanical properties. Young's modulus is the ratio between normal stress and longitudinal strain within the elastic limit [25]. To measure the tensile strength and elongation at break of each membrane, we used a universal testing machine (UTM) (FH, Tinius Olsen). The tensile strength and elongation of a membrane depends on its morphological structure and porosity. Membranes with large voids are unsuitable because they have regions lacking integrity (weak points). Under a high operating pressure, this type of membrane may fail. Therefore, the hydrophilic membrane was more stable than the hydrophobic membrane. Figure 4 shows the mechanical stability of the hydrophilic and hydrophobic membranes.

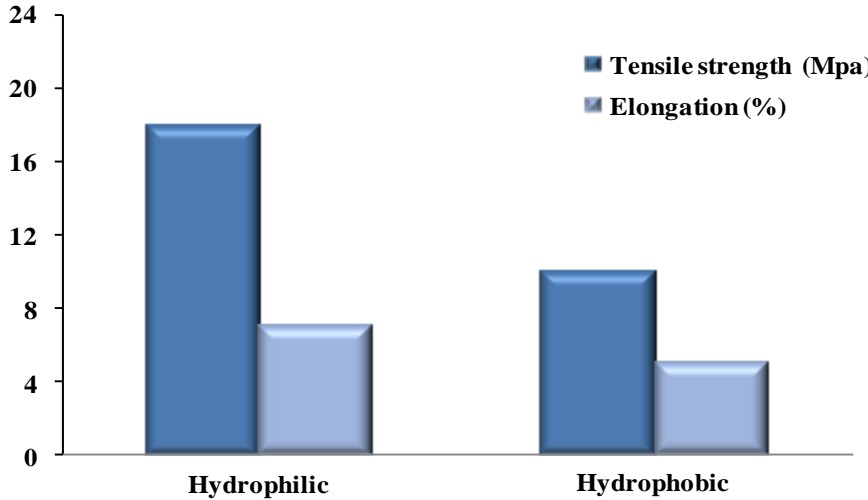

**Figure 4.** Tensile strength (Mpa) and elongation (%) for hydrophobic and hydrophilic membranes.

The spreading of a liquid on a solid surface is known as wettability. Wettability for both hydrophobic and hydrophilic membranes depends on contact angles and surface tension. In addition to the feed solution, it is dependent on the morphology of the solid surface, chemical composition, pore size, and membrane treatment. Whenever the contact angle is small (less than 90°), the membrane's behavior will be hydrophilic. With a large contact angle (more than 90°), the membrane will show hydrophobic behavior. The hydrophobic membrane's behavior was observed to exhibit low wettability at high contact angles (above 90°) due to low surface energy. However, the hydrophilic membrane's behavior was observed to exhibit high wettability at low contact angles (below 90°) due to high surface

energy. The aim of this research was to evaluate the performance of the membranes based on SEM technology, mechanical stability, and other factors. Therefore, the measurements of the wettability and contact angles were not taken into consideration for this study. However, the factors taken into account were sufficient to give an indication of the performance of each membrane.

### 3.2. Effect of Pressure

The results of pressure experiments line up with the results of flux experiments and their discussion. Figures 5 and 6 show the results of flux with time at 1, 1.5, and 2 bar of pressure for distilled water and phenol solutions using hydrophilic and hydrophobic membranes to assess pressure. Flux increased with pressure for distilled water and the phenol solution at 1, 1.5, and 2 bars. The results of the phenol solution's flux after 40 min for the hydrophilic membrane were 35.47, 42.44, and 53.44 L/m$^2$ h, respectively; and for the hydrophobic membrane were 40.72, 47.35, and 58.44 L/m$^2$ h, respectively. Fluxes for distilled water after 40 min were 57.89, 62.88, and 68.98 L/m$^2$ h, respectively, for the hydrophilic membrane; and for the hydrophobic membrane were 77.40, 83.40, and 88.89 L/m$^2$ h, respectively. This can be related to the effect of pressure [26] (when pressure is increased, more feed solution is forced through the membrane). However, with time, the flux was slightly decreased as a result of membrane fouling. Mohammad et al. [27] mentioned that the flux rate decreased with time for hydrophilic and hydrophobic membranes; this could have been due to fouling of the membrane. Benitez et al. [28] mentioned that an increase in water flux causes more permeate to permeate by increasing the pressure. Additionally, Arsuaga et al. [29] declared that the permeation flux increased as the transmembrane pressure increased, up to 1.66 MPa. Although in hydrophilic membranes, the addition of PEO to the casting solution increased the water permeation, with time the membrane's pores became clogged, which caused a decrease in water flux [18,30]. Additionally, Figures 5 and 6 show that the distilled water's flux, more than the phenol solution's flux, depends on the volume of the particles in the solution. This could be explained by the fact that the particle size of the phenol solution was larger than the particle size of the distilled water. As a result, particles larger than the membrane's pores would clog the membrane's surface, resulting in a decrease in flux with time [31–33].

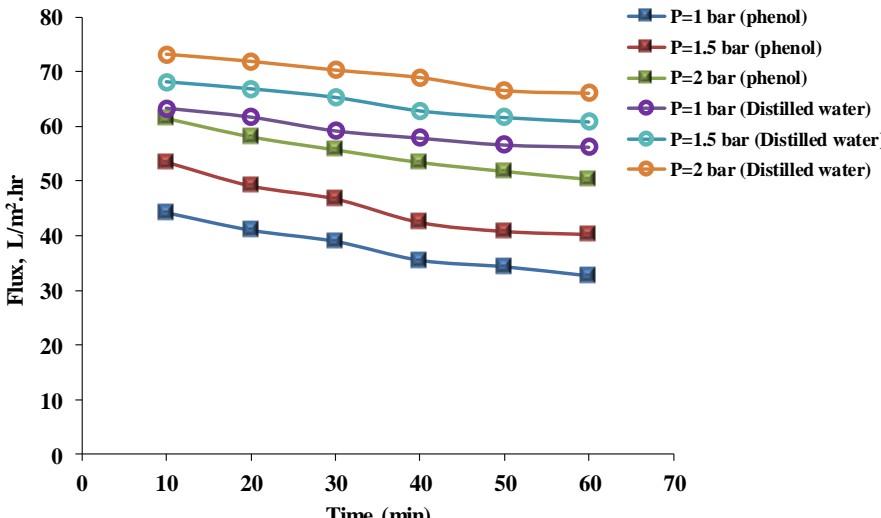

**Figure 5.** The phenol solution and distilled water's fluxes with time for a hydrophilic membrane at different pressures (phenol concentration = 50 mg/L, temperature = 30 °C).

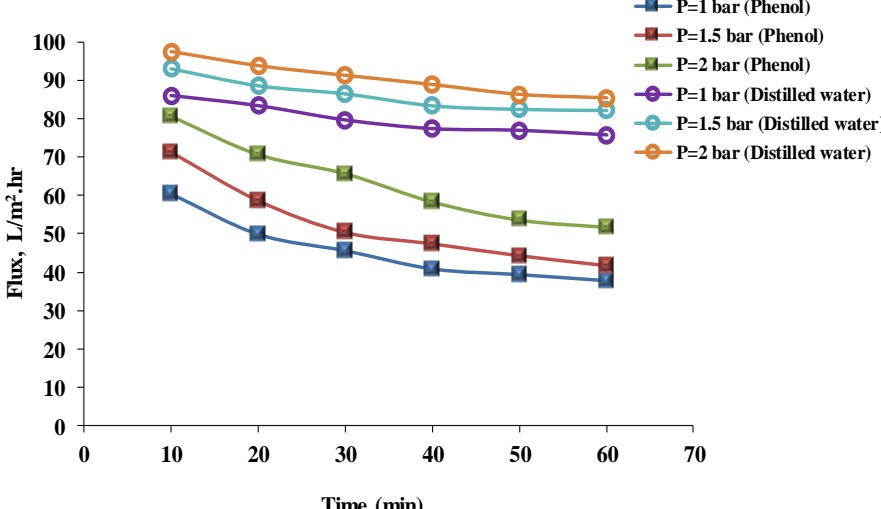

**Figure 6.** Phenol solution and distilled water's fluxes with time for a hydrophilic membrane at different pressures (phenol concentration = 50 mg/L, temperature = 30 °C).

There are many studies on the influence of pressure on membrane performance. Mnif et al. [34] reported that a slight increase in phenol retention with an increase in pressure would lead to a more diluted permeate and increased water flux. The microfiltration in this study made little difference in the results due to using different materials and mixing rates. Abu-Dalo et al. [35] established a relationship among the additive amount, the membrane's morphology, and the flux after a polymer membrane was mixed with carbon nanotubes (CNTs) in different mixing rates. Their results were a higher water flux of 69.71 kg/h m$^2$ at 1 bar using 1% by weight of carbon nanotubes, but water flux of about 37.8 kg/h m$^2$ at a lower percentage of mixing weight (less than 0.5%). The rejection increased by 0.5 percent by weight and decreased by one percent by weight.

*3.3. Effect of Concentration*

To illustrate, the water flux decreased over time as the phenol solution's concentration increased from 10 to 80 mg/L when hydrophilic and then hydrophobic membranes were used. Figures 7 and 8 show that the fluxes at phenol solution concentrations of 10, 50, and 80 mg/L were 39.66, 35.47, and 28.38 L/m$^2$ h, respectively, for the hydrophilic membrane. For the hydrophobic membrane, the fluxes were 47.47, 40.72, and 34.38 L/m$^2$ h, respectively, after 40 min. The cause of these results was the increase in phenol concentration. The particles in the solution increased and caused the pores to clog. Additionally, if the particle concentration in the solution is high, the particles will be accumulated on the membrane surface, and when phenol concentration is increased [30], the number of particles is increased, forming a thin layer on the membrane and preventing the phenol solution from permeating, but such a drop was not observed with a flux of distilled water [33,36]. The previous findings may be consistent with those of Benosmane et al. [21], who found that increasing the phenol concentration increased the permeation flux from 0 to 10$^{-3}$ m. Beyond that, the permeation flux decreased due to the formation of a milky layer on the membrane's surface. Furthermore, the flux was affected by the feed solution's concentration, as demonstrated by Yahya et al. [37]. When nanofiltration membranes were prepared with PES and polyphenylsulfone (PPSU) to remove 4-nitrophenol (4-NP), the flux decreased from 4.11 to 3.20 L/m$^2$/h at 3 bar when the concentration increased from 10$^{-5}$ to 10$^{-3}$ mol/L.

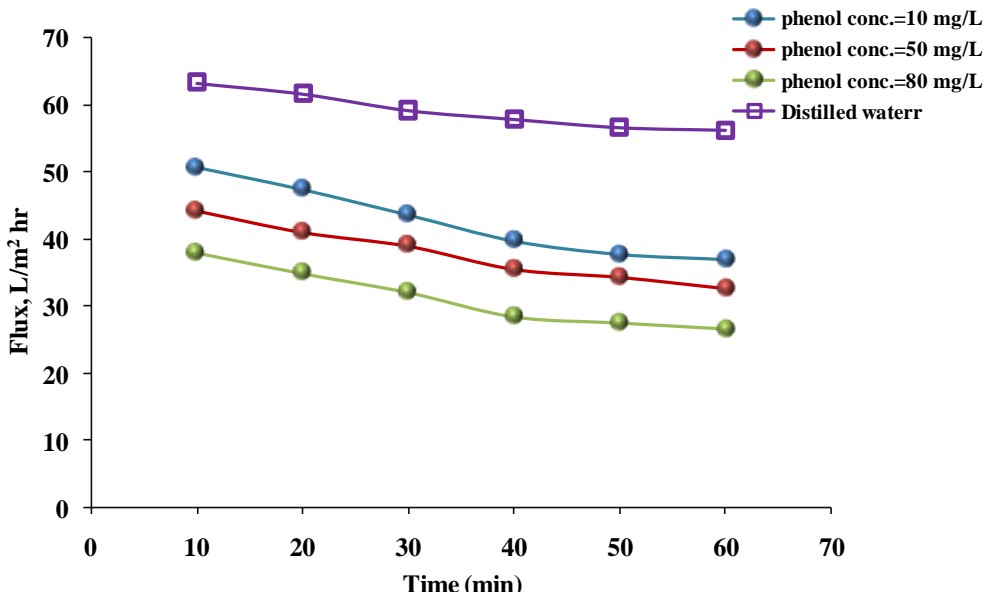

**Figure 7.** Phenol solution and distilled water's fluxes with time for a hydrophilic membrane at different concentrations (pressure = 1 bar, temperature = 30 °C).

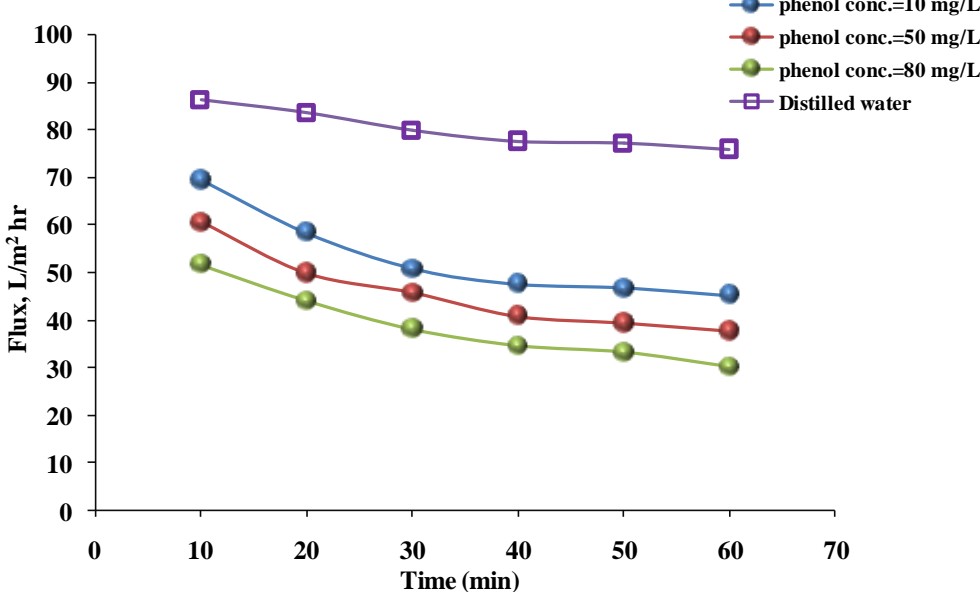

**Figure 8.** Phenol solution and distilled water's flux with time for a hydrophobic membrane at different concentrations (pressure = 1 bar, temperature = 30 °C).

### 3.4. Effect of Temperature

The flux results for distilled water and the phenol solution at different temperatures (10, 30, and 50 °C) are revealed in Figures 9 and 10. The results for the phenol solution's flux were 30.84, 35.47, and 39.21 L/m$^2$ h, respectively, for the hydrophilic membrane; and 33.79, 40.72, and 48.64 L/m$^2$ h, respectively, for the hydrophobic membrane after 40 min. Flux values for distilled water after 40 min were 53.38, 57.89, and 62.87 L/m$^2$ h, respectively, for the hydrophilic membranes; and for the hydrophobic membrane were 74.56, 77.40, and 84.43 L/m$^2$ h, respectively. As the temperature increases, the water flux also increases. The viscosity of the liquid decreases with an increase in temperature, and as a result, the liquid permeation through the membrane increases. Furthermore, the temperature increase simplifies the dispersion process and increases the solution's solubility. However, the flux

still decreases with time because the membrane has a certain heat resistance [26]. Much research has been published on the effect of temperature on the permeate flux of phenol solutions. The results obtained when the effect of temperature on the performance of ultrafiltration was studied are similar to others in the current research. Increasing the temperature of the solution led to an increase in flux. However, the maximum flux was recorded when the temperature did not exceed 50 °C [38]. Mänttäri et al. [39] investigated the effects of temperature on membrane flux and retention, and found that the maximum flux was at 65 °C, after which the flux decreased. Unless the membranes were pre-treated, the results were identical to those of this study. A few membranes can endure a temperature of 65 °C without pre-treatment; otherwise, membrane efficiency is affected.

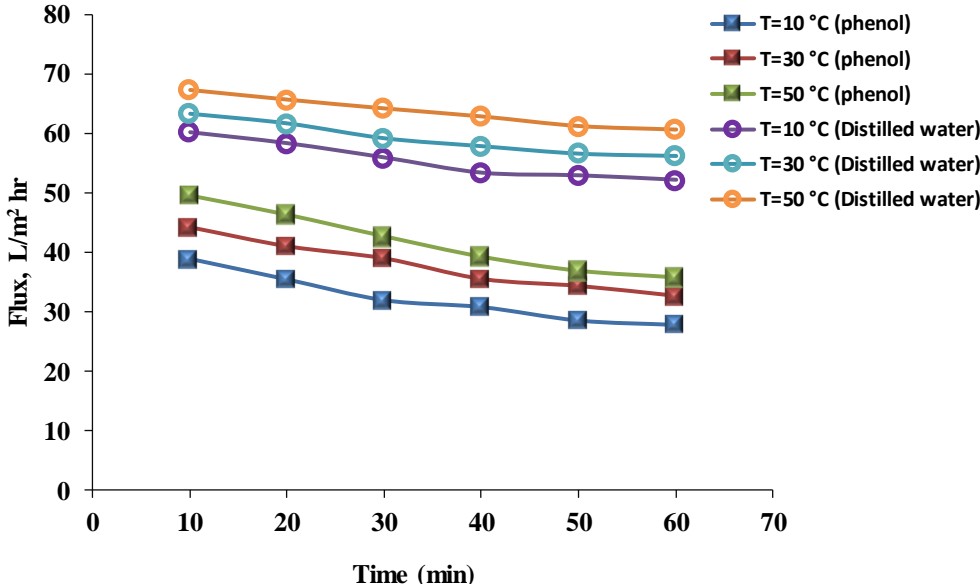

**Figure 9.** Phenol solution and distilled water's fluxes with time using a hydrophilic membrane at different temperatures (pressure = 1 bar, phenol concentration = 50 mg/L).

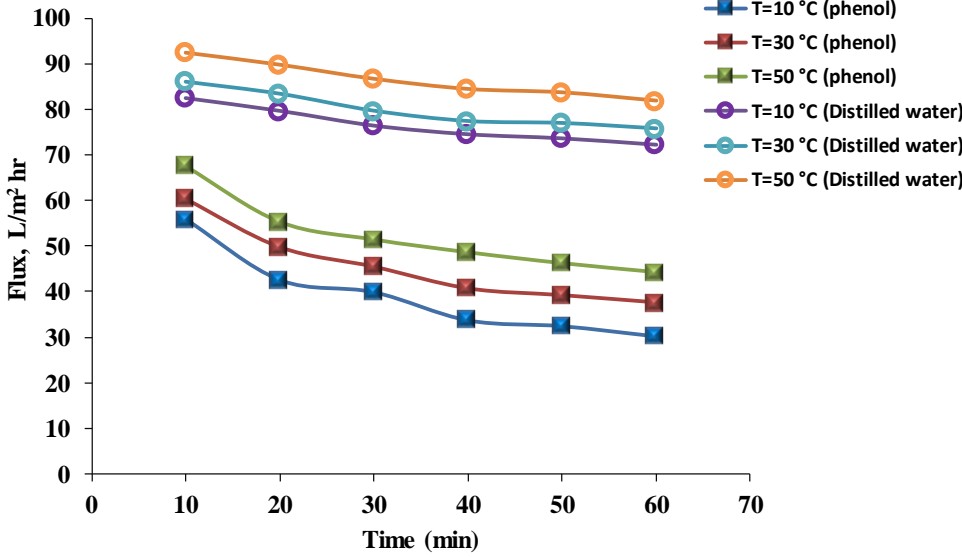

**Figure 10.** Phenol solution and distilled water's fluxes with time using a hydrophobic membrane at different temperatures (pressure = 1 bar, phenol concentration = 50 mg/L).

### 3.5. Comparison of Hydrophilic and Hydrophobic Membranes

The comparison of the percentage of phenol rejection for hydrophilic and hydrophobic membranes with time is shown in Figure 11. It illustrates greater rejection for the hydrophilic membrane (60% for the phenol solution) compared with the hydrophobic membrane (45% for the phenol solution). Especially during the initial period, the hydrophobic membrane's decreasing rejection trend was greater than that of the hydrophilic membrane. Meanwhile, rejection for both membranes steadily decreased. This is due to the clogging of the membrane's surface, which leads to a decrease in the transfer of water permeating across the membrane with time [33]. Rezaee et al. [40] showed that the rejection of arsenate by the PSF/GO membrane could be affected by various parameters, such as pH, initial concentration, and pressure. Rajesha et al. [41] described that the rejection was affected by the pollutant concentration when cellulose acetate/zinc oxide/zeolite composite membranes were used to remove benzophenone-3 from water. At first, the concentration increased the rejection to 98% and supported the membrane's hydrophilicity, but the rejection decreased with respect to the benzophenone-3 concentration. When the permeate concentration (benzophenone-3) increased, the rejection increased, whereas the rejection decreased with higher pressure (10 bar).

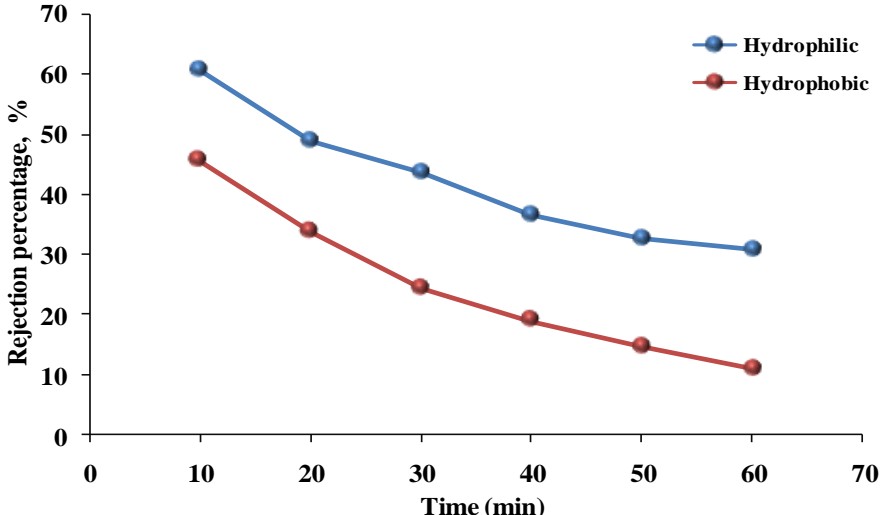

**Figure 11.** Phenol solution's rejection percentage versus time for hydrophilic and hydrophobic membranes (pressure = 1 bar, temperature = 30 °C, phenol concentration = 50 mg/L).

Figure 12 illustrates the variation in permeate flux with time for both hydrophilic and hydrophobic membranes using a phenol solution. The decrease in flux through the hydrophilic membrane with time was steady and constant; this could have been due to membrane fouling. While adding PEO to the casting solution increased water permeation, the membrane pores became clogged over time, resulting in a decrease in permeate flux [18]. The flux through the hydrophobic membrane was higher than that through the hydrophilic membrane, but the rate of flux through the hydrophobic membrane decreased greatly, particularly during the initial period, due to membrane fouling. On the other hand, in spite of the PEO additive being used to ease the permeation, the pore size of the hydrophilic membrane was smaller than that of the hydrophobic membrane. This could have been due to PEO molecules filling up voids between PSF molecules [25].

Milescu et al. [15] showed a membrane developed from PES using the bio-based solvent Cyrene and compared it to PES produced using the traditional N-methyl pyrrolidinone (NMP) and PVP. As an additive, porosity and pore size distribution were studied at multi-weight mixing using hot (70 °C) and cold (17 °C) casting gels of PES and PVP. The outcome was 79% porosity when Cyrene-based membranes were used. This is a higher result for porosity than NMP, despite the fact that no additives (pore-forming agents) were

used. However, an NMP-based membrane using pore forming (PVP) resulted in 76.70% porosity, and the result decreased to 54.90% when PVP was increased in terms of weight percent. The greatest flux for a microfiltration membrane was found for 76.90% Cyrene, 7.70% PVP, and 15.50% PES. Aminudin et al. [18] reported the effects of the additive materials, the composition of the membrane, and the additive amount. By increasing additive PEG from 0.5% to 5.0% by weight and additive PVP from 0.5% to 5.0% by weight, the water permeates increased from 14.73 to 101.85 LMH and from 21.13 to 177.61 LMH, respectively. The hydrophobic membrane was also tested with PSF in the current study. It had a higher flux than the hydrophilic membrane because the pore size of the hydrophobic membrane was larger than that of the hydrophilic membrane.

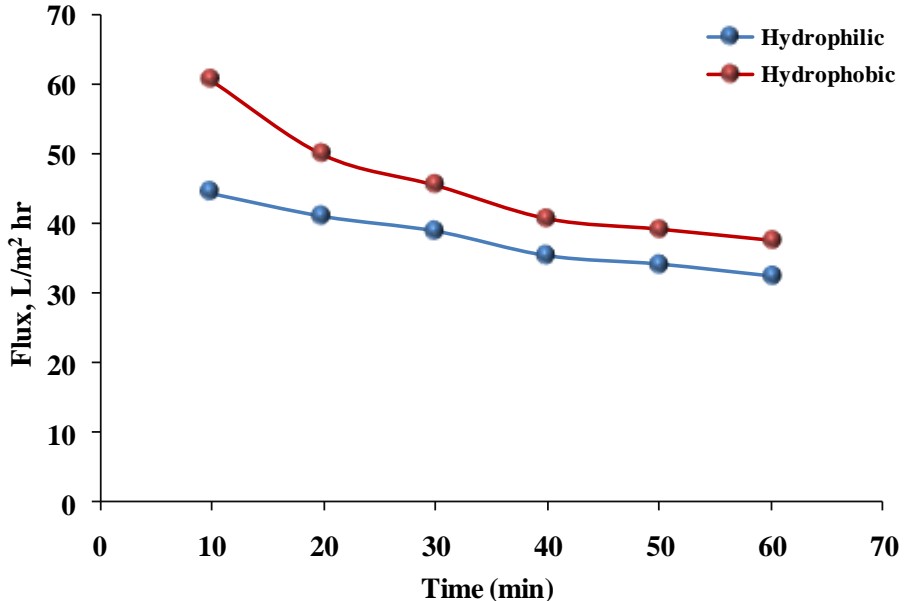

**Figure 12.** Flux versus time of the phenol solution for hydrophilic and hydrophobic membranes (pressure = 1 bar, temperature = 30 °C, phenol concentration = 50 mg/L).

## 4. Conclusions

Hydrophilic and hydrophobic microfiltration membranes were characterized in this study. The feasibility of the removal of phenol was systematically investigated with various pressures, concentrations, and temperatures. Distilled water and a phenol solution were used. The hydrophilic membrane was made using PEO in a greater ratio than NMP; thus, it had a low water-flux value and a high percentage of rejection. The maximum flux was found for the phenol solution after 40 min at 2 bars of pressure, with a phenol concentration of 10 mg/L, and a temperature of 50 °C. The maximum flux for the hydrophobic membrane was higher than that of the hydrophilic membrane. When the process was enhanced by variable pressures at the phenol concentration of 50 mg/L and the temperature of 30 °C, after 40 min, the maximum fluxes were increased by 51% and 43% for hydrophilic and hydrophobic membranes, respectively. Furthermore, the flux for the phenol solutions with various concentrations, after 40 min, at 2 bars of pressure and 50 °C, showed that the flux results decreased by 28% and 27% for hydrophilic and hydrophobic membranes, respectively, as the concentration increased. When the work depended on a variable temperature, after 40 min, at 2 bars, and with a phenol concentration of 50 mg/L, the flux results were increased by 27% and 44% for hydrophilic and hydrophobic membranes, respectively. The same behavior was observed in distilled water, though with greater differences than the phenol solutions. It can also be concluded that the flux was reduced with operation time in different conditions. The operating pressure has a stronger effect on the system, and the permeation flux can be enhanced with an increase in operating pressure. The increase in the rejection percentage was greater for the hydrophilic membrane (60%)

compared with the hydrophobic membrane (45%). As a result, PSF with the additive PEO and solvent NMP has a significant performance advantage based on the results of rejection and flux. The results show that the method has a strong ability to remove phenol from water samples. It could be applied to real wastewater.

**Author Contributions:** Conceptualization, T.K.H. and A.-S.T.A.-M.; data curation, T.K.H. and N.A.J.; formal analysis, A.-S.T.A.-M.; investigation, T.K.H. and A.-S.T.A.-M.; methodology, T.K.H.; resources, N.A.J.; software, N.A.J.; supervision, A.-S.T.A.-M.; validation, N.A.J.; writing—original draft, T.K.H. All authors have read and agreed to the published version of the manuscript.

**Funding:** This research received no external funding.

**Institutional Review Board Statement:** Not applicable.

**Data Availability Statement:** Not applicable.

**Acknowledgments:** The authors would like to thank the Mustansiriyah university (www.uomustansiriyah.edu.iq; 21 March 2023) Baghdad—Iraq, for its support in the present work.

**Conflicts of Interest:** The authors declare no conflict of interest.

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
