# Peer review of "The Performance of Microfiltration Using Hydrophilic and Hydrophobic Membranes for Phenol Extraction from a Water Solution"

_2305-7084, doi:10.3390/chemengineering7020026_

Round 1

Reviewer 1 Report

In this work, hydrophilic and hydrophobic polysulfone-based membrane were fabricated via the phase inversion technique for phenol extraction from wastewater. The results of this study provides the reference for phenol treatment. However, some parts appear to be not prepared well. Therefore, I recommend publishing this work after addressing the following comments:

1. In the Results and Discussion, please demonstrate the wettability of the synthesized membranes.

2. Please explain the effect of wettability on the separation performance.

3. The investigation for stability of two-types of membranes should be supplemented, and gives the reason for the difference on the stability of hydrophilic and hydrophobic membranes.

4. This paper has some mistakes. For instance, the number of significant digits should be consistent; An explanation of the abbreviation only needs to be provided when it appears for the first time; The caption is wrong. For example, Figure 10. etc. The authors should get the paper proofread carefully before publishing it.

Author Response

ChemEngineering-2209579– Revised Manuscript

RC = Reviewer’s Comment; AR = Authors’ Response

Reviewer #1:

RC: “In this work, hydrophilic and hydrophobic polysulfone-based membrane were fabricated via the phase inversion technique for phenol extraction from wastewater. The results of this study provides the reference for phenol treatment. However, some parts appear to be not prepared well. Therefore, I recommend publishing this work after addressing the following comments:

AR: The authors made the required changes as suggested by the reviewers (see the revised manuscript). The authors clarified the comments raised by the reviewer as stated below.

RC: “1. In the Results and Discussion, please demonstrate the wettability of the synthesized membranes.”

AR: The authors appreciate the reviewer suggestion. The authors demonstrated the wettability of the synthesized membranes as follows:

“The spreading of a liquid on a solid surface is known as wettability. Wettability for both hydrophobic and hydrophilic membranes depends on contact angles and surface tension. In addition to the feed solution, it is dependent on the morphology of the solid surface, chemical composition, pore size, and membrane treatment. Whenever the contact angle is small (less than 90°), the membrane behavior is hydrophilic. With a large contact angle (more than 90°), the membrane shows hydrophobic behavior. The hydrophobic membrane behavior was observed to exhibit low wettability at high contact angles (above 90°) due to low surface energy. However, the hydrophilic membrane behavior was observed to exhibit high wettability at low contact angles (below 90°) due to high surface energy. The aim of this research was to evaluate the performance of the membrane based on SEM technology, mechanical stability, and other factors. Therefore, the measurements of the wettability and contact angles were not taken into consideration for this study. However, the factors taken into account were sufficient to give an indication of the performance of the membranes."

RC: “2. Please explain the effect of wettability on the separation performance.”

AR: The authors explained the effect of wettability on the separation performance as follows:

" Whenever the contact angle is small (less than 90°), the membrane behavior is hydrophilic. With a large contact angle (more than 90°), the membrane shows hydrophobic behavior. The hydrophobic membrane behavior was observed to exhibit low wettability at high contact angles (above 90°) due to low surface energy. However, the hydrophilic membrane behavior was observed to exhibit high wettability at low contact angles (below 90°) due to high surface energy."

RC: “3. The investigation for stability of two-types of membranes should be supplemented, and gives the reason for the difference on the stability of hydrophilic and hydrophobic membranes.“

AR: The authors added a new paragraph in the “results and discussion” part and presented new figure about the difference in the stability of hydrophilic and hydrophobic membranes as follows (see the revised manuscript):

"The mechanical stability of membranes was assessed by determination of Young’s modulus and elongation at the break of prepared membranes. It is essential to study the mechanical stability to evaluate the lifetime of membranes that could be obtained by examining the mechanical properties. Young’s modulus is the ratio between normal stress and longitudinal strain within the elastic limit [25]. To measure the tensile strength and percentage elongation of membranes using a universal testing machine (UTM) (FH, Tinius Olsen). The tensile strength and elongation of the prepared membranes depend on the morphological structure, and porosity of the membrane. Membranes with large voids were unsuitable because they display lacking intensity regions (weak points), thus under high operating pressure this type of membrane may fail. Therefore, a hydrophilic membrane is more stable than a hydrophobic membrane. Figure 4 shows the mechanical stability of hydrophilic and hydrophobic membranes.”

RC: 4. This paper has some mistakes. For instance, the number of significant digits should be consistent; An explanation of the abbreviation only needs to be provided when it appears for the first time; The caption is wrong. For example, Figure 10. etc. The authors should get the paper proofread carefully before publishing it. “

AR: The authors adjusted the number of significant digits, abbreviations, and figures captions as suggested. The authors Proofread the revised paper.

Reviewer 2 Report

This paper investigated the effect of membranes structure and properties on phenol extraction from wastewater Some issues should be addressed.

1.       The title should be modified. The description of preparation and characterization of membranes is less in the results and discussion section. In other words, these are not key points in this work.

2.       How to evaluate the hydrophobicity and hydrophilicity of membranes, such as water contact angle. Such data are important.

3.       Can author analyze more the microstructure. How about the pore size distribution.

4.       What is the mechanism for the superior water flux for the hydrophobic type, whereas better rejection for the hydrophilic type. The key point should be presented in Abstract section.

5.       Comment: More relevant works should be cited for enriching the introduction section, such as other-type membrane. (International Journal of Biological Macromolecules, https://doi.org/10.1016/j.ijbiomac.2022.12.052; Carbohydrate Polymers, https://doi.org/10.1016/j.carbpol.2022.119601)

Author Response

ChemEngineering-2209579– Revised Manuscript

RC = Reviewer’s Comment; AR = Authors’ Response

Reviewer #2:

RC: “This paper investigated the effect of membranes structure and properties on phenol extraction from wastewater Some issues should be addressed.”

AR: The authors made the required changes as suggested by the reviewers (see the revised manuscript). The authors clarified the comments raised by the reviewer as stated below.

RC: “1.       The title should be modified. The description of preparation and characterization of membranes is less in the results and discussion section. In other words, these are not key points in this work. “

AR: The authors agreed to the reviewer's comment and adjusted the title as follows:

"The Performance of Microfiltration using Hydrophilic and Hydrophobic Membranes for Phenol Extraction from a Water Solution"   

RC: “2.       How to evaluate the hydrophobicity and hydrophilicity of membranes, such as water contact angle. Such data are important. “

AR: The authors provided more information on contact angles as follows:

“The spreading of a liquid on a solid surface is known as wettability. Wettability for both hydrophobic and hydrophilic membranes depends on contact angles and surface tension. In addition to the feed solution, it is dependent on the morphology of the solid surface, chemical composition, pore size, and membrane treatment. Whenever the contact angle is small (less than 90°), the membrane behavior is hydrophilic. With a large contact angle (more than 90°), the membrane shows hydrophobic behavior. The hydrophobic membrane behavior was observed to exhibit low wettability at high contact angles (above 90°) due to low surface energy. However, the hydrophilic membrane behavior was observed to exhibit high wettability at low contact angles (below 90°) due to high surface energy. The aim of this research was to evaluate the performance of the membrane based on SEM technology, mechanical stability, and other factors. Therefore, the measurements of the wettability and contact angles were not taken into consideration for this study. However, the factors taken into account were sufficient to give an indication of the performance of the membranes.”

RC: “3.       Can author analyze more the microstructure. How about the pore size distribution.”

AR: The authors added a new paragraph in the “results and discussion” part to provide analysis on the microstructure and pore size distribution (see the revised manuscript):

 “The SEM was used to determine the average pore size and pore size distribution. The top surface, the bottom surface, and the section of the cross-surface membrane were fractured cryogenically in liquid nitrogen to leave an un-deformed structure. Then, it was attached to sample stubs with double-surface gold using a sputter coater. After that, the samples were imaged using SEM. The image analysis was carried out to obtain the average pore size. From Figure 2a, the average pore size was 521 nm for a hydrophobic membrane. From Figure 3a, the average pore size for the hydrophilic membrane was 335 nm.”

RC: “4.       What is the mechanism for the superior water flux for the hydrophobic type, whereas better rejection for the hydrophilic type. The key point should be presented in Abstract section.“

AR: The authors presented the key point in the Abstract as follows:         

“The hydrophobic membrane showed high fluxes and low rejection. So, transport through this membrane is closer to viscous behaviour than for the hydrophilic membrane; in contrast, the permeability through the hydrophilic membrane is less because the pore size decreases the viscous flow mechanism.”

RC: “5.       Comment: More relevant works should be cited for enriching the introduction section, such as other-type membrane. (International Journal of Biological Macromolecules, https://doi.org/10.1016/j.ijbiomac.2022.12.052; Carbohydrate Polymers, https://doi.org/10.1016/j.carbpol.2022.119601)”

AR: The authors cited both references in the Introduction section.

Reviewer 3 Report

The article presents simple experimental research carried out on a water phenol solution. The authors should remember, that technological sewage has a very diverse quality composition and should emphasize this fact in the article.

The authors should take into account the following comments: In the title of the article, replace "... from Wastewater", to: "... from a water solution".

L47-48- I don't understand these informations

L54-55: Remove the repetition for: acetone

L75-80 - Emphasize the novelty of conducted research

L89 - Correct the unit of conductivity

L102-105- Improve style

Fig. 1. - Improve the quality of the figure (descriptions of the filter system elements)

L151 - What was the analytical method used before the phenol measurement on the spectrophotometer?

L167-169 - Information on the SEM measurement method should be transferred to Section 2.

L279- Correct the concentration value at: 10-5 or 0.00001 mol/L; and a unit for 0.001 concentration to: mol/L

L388 - correct the description of the drawing (Hydrophobic is twice) Section 4 - Conclusions must be rewritten - they cannot be a re -description of the results.

Author Response

ChemEngineering-2209579– Revised Manuscript

RC = Reviewer’s Comment; AR = Authors’ Response

Reviewer #3:

RC: “The article presents simple experimental research carried out on a water phenol solution. The authors should remember, that technological sewage has a very diverse quality composition and should emphasize this fact in the article.

The authors should take into account the following comments:”

AR: The authors made the required changes as suggested by the reviewers (see the revised manuscript). The authors clarified the comments raised by a reviewer as stated below.

RC: “In the title of the article, replace "... from Wastewater", to: "... from a water solution".“

AR: The authors agreed to the reviewer's comment and adjusted the title as follows:

"The Performance of Microfiltration using Hydrophilic and Hydrophobic Membranes for Phenol Extraction from a Water Solution"   

RC: L47-48- I don't understand these informations”

AR: The authors agreed with reviewer's comment and removed this information.

RC: “L54-55: Remove the repetition for: acetone“

AR: The authors adjusted that suggestion.

RC: “L75-80 - Emphasize the novelty of conducted research“

AR:  The authors presented the novelty of this research as follows:

“In Iraq, there are many petroleum refining industries, and phenol is one of their most dangerous wastes. Therefore, the importance of the study is to remove phenol using a membrane made of polysulfone polymer.”

RC: “L89 - Correct the unit of conductivity“

AR:  Adjusted as suggested.

RC: “L102-105- Improve style“

AR: The authors improved the sentence in the text as follows:

“The PEO was used as an additive. The NMP was mixed with polysulfone as a solvent in hydrophilic membranes.”

RC: “Fig. 1. - Improve the quality of the figure (descriptions of the filter system elements)“

AR: The authors improved Figure 1 as suggested.

RC: L151 - What was the analytical method used before the phenol measurement on the spectrophotometer?“

AR: The authors provided more information as follows: 

" The content of phenol was determined by the Folin-Ciocalteu method as described in Jasim and Hussein's [20] study. The absorbance was measured by a UV spectrophotometer (Shimadzu 1800, Japan) at a wavelength of 270 nm. The absorbance recorded from the UV spectrophotometer was used to prepare a standard calibration curve to determine phenol permeate concentration."   

RC: “L167-169 - Information on the SEM measurement method should be transferred to Section 2.“

AR: Adjusted as suggested.

RC: “L279- Correct the concentration value at: 10-5 or 0.00001 mol/L; and a unit for 0.001 concentration to: mol/L.“

AR: Adjusted as suggested.

RC: “L388 - correct the description of the drawing (Hydrophobic is twice) “

AR: Adjusted as suggested.

RC: “Section 4 - Conclusions must be rewritten - they cannot be a re -description of the results.“

AR: The authors have re-written the conclusion as follows:

“Hydrophilic and hydrophobic microfiltration membranes were characterized in this study. The feasibility of the removal of phenol was systematically investigated using variable conditions such as pressure, concentration, and temperature. Distilled water and a phenol solution were used. The hydrophilic membrane was made using PEO in a greater ratio than NMP; thus, it had a low water flux and a high percentage of rejection. The maximum flux results for phenol solution after 40 min at pressure 2 bars, phenol concentration 10 mg/l, and temperature 50 °C. The maximum flux for the hydrophobic membrane showed a higher result rate than the hydrophilic membrane. When the work was enhanced by variable pressures at phenol concentrations of 50 mg/l and temperatures of 30 °C after 40 minutes, the maximum fluxes were increased by 51% and 43% for hydrophilic and hydrophobic membranes, respectively. Furthermore, the maximum flux result for phenol solution after 40 min at 2 bars of pressure and 50 °C with variable concentrations showed that the flux results decreased by 28% and 27% for hydrophilic and hydrophobic membranes, respectively, in series with concentration increasing. When the work depends on a variable temperature after 40 min, a pressure of 2 bars, and a phenol concentration of 50 mg/l, the flux results in increases of 27% and 44% for hydrophilic and hydrophobic membranes, respectively. The same behavior was observed in distilled water with a higher rate of a difference than in phenol solution. It can also be concluded that the flux is reduced with operation time at different conditions. The operating pressure has a stronger effect on the system, and permeation flux can be enhanced with the increase in operating pressure. The increase in the rejection percentage was greater for the hydrophilic membrane (60%) compared with the hydrophobic membrane (45%). As a result, PSF with the used additive PEO and solvent NMP has a significant performance based on the result of rejection and flux. The results show that the method has a high ability to remove phenol from water samples. It could be applied to real wastewater.”

Round 2

Reviewer 1 Report

This article could be accepted in present form.

Reviewer 2 Report

This paper can be accepted in that form.